# Biological Therapy of Severe Asthma and Nasal Polyps

**DOI:** 10.3390/jpm12060976

**Published:** 2022-06-16

**Authors:** Agamemnon Bakakos, Florence Schleich, Petros Bakakos

**Affiliations:** 1Department of Respiratory Medicine, National and Kapodistrian University of Athens, 157 72 Athens, Greece; petros44@hotmail.com; 2Department of Respiratory Medicine, University of Liege, GIGA I3, 4000 Liege, Belgium; fschleich@chuliege.be

**Keywords:** asthma, chronic rhinosinusitis with nasal polyps (CRSwNP), mepolizumab, benralizumab, dupilumab, monoclonal antibodies

## Abstract

Chronic rhinosinusitis is a common disease worldwide and can be categorized into chronic rhinosinusitis with nasal polyps and chronic rhinosinusitis without nasal polyps. Chronic rhinosinusitis with nasal polyps is common in patients with asthma and, particularly, severe asthma. Severe asthma is effectively treated with biologics and the coexistence of severe asthma with chronic rhinosinusitis with nasal polyps presents a phenotype that is more likely to respond to such treatment. In this review, we focus on the link between asthma and nasal polyps, and we review the treatment effect of various monoclonal antibodies in patients with severe asthma and nasal polyps as well as in patients with nasal polyps without asthma or with mild-to-moderate asthma. With the enhancement of our armamentarium with new monoclonal antibodies the right choice of biologic becomes an important target and one that is difficult to achieve due to the lack of comparative head-to-head studies.

## 1. Introduction

The prevalence of chronic rhinosinusitis is around 10% in the US and Europe [1]. Chronic rhinosinusitis can be categorized into two phenotypes: chronic rhinosinusitis with nasal polyps (CRSwNP) and chronic rhinosinusitis without nasal polyps (CRSnNP) [2]. CRSwNP is a chronic inflammatory disease of the nasal passage linings or sinuses leading to soft tissue growth, known as nasal polyps, in the upper nasal cavity [3]. Symptoms experienced by patients with nasal polyposis include nasal blockage, loss of smell, rhinorrhea, sleeping difficulties and impairments in social, emotional and lifestyle well-being but also symptoms derived from lower airway involvement [4,5].

Current treatment options for patients with nasal polyposis (NP) are intranasal and, for severe cases, oral corticosteroids (OCS) and long-term antibiotics [3,6]. Surgery to remove the polyp tissue may also be indicated for severe cases [7]. However, polyps have a strong tendency to recur, often leading to the need for repeated surgery. Recently, monoclonal antibodies (mAbs) have been implemented in the treatment of difficult-to-treat CRSwNP, targeting specific molecules that are implicated in the inflammatory process, in an effort to relieve patients, not only from their symptoms but also, from the burden of OCS and their numerous side effects. The biologics that have been approved are mepolizumab, benralizumab and dupilumab [8].

Patients with CRSwNP have a Th2-predominant type of inflammation and present high levels of interleukin-5 (IL-5) and eosinophilic inflammation, an image that is often also seen in severe asthma, where the aforementioned mAbs are also used with great efficacy [9,10]. However, NP is not characterized by high production of IL-5 and eosinophilia in all patients. Asthma, and especially severe asthma, is associated with various comorbidities and NP is one of the most frequent [11,12]. Asthma affects around 4.3% of people worldwide, with its prevalence reaching up to 10% in the United States [13]. Although most asthmatics have a benign level of the disease, around 5% suffer from severe asthma, with many exacerbations, impaired lung function, worse quality of life and repeated use of OCS [14]. The most common endotype of severe is asthma is the Th2 high type, with eosinophils being pivotal in the development of airway inflammation [15]. Chronic rhinosinusitis with nasal polyposis affects around 40% to 60% of severe eosinophilic asthma [16,17,18,19]. Therefore, it is of great interest to further elucidate the link between severe asthma and CRSwNP, especially since their coexistence can be managed with the use of the same mAbs. Accordingly, the question that arises is which biologic to choose and based on which data or criteria. Our aim in this review is to summarize current knowledge on this topic.

## 2. The Link between Severe Asthma and Nasal Polyps

The association between asthma and CRSwNP is not just a simple coexistence. Core underlying pathophysiological mechanisms are shared between them, with T2 inflammation being the cornerstone of both airway disorders. Thus, taking into account that the extent of T2 inflammation strongly impacts the severity of both diseases, it can be expected that patients suffering from severe asthma will usually experience severe CRSwNP symptoms too, and vice versa [20]. Around one out of two patients with CRSwNP have some form of lower airway disease, and the more severe asthma becomes, the more likely is a patient will have nasal polyposis [21].

T2-high inflammation results from two different molecular pathways, eventually leading to the production of cytokines such as interleukin-4 (IL-4), interleukin-5 (IL-5) and interleukin-13 (IL-13), which in turn orchestrate the production, migration and tissue infiltration of eosinophils and mast cells. The first pathway revolves around the adaptive immune response through the stimulation of T-helper 2 (T_H_2) CD4^+^ cells by antigen presentation. The second pathway revolves around the innate immune response since the type 2 innate lymphoid cells (ILC2) are triggered by alarmins such as thymic stromal lymphopoietin (TSLP), interleukin-25 (IL-25) and interleukin-33 (IL-33). These alarmins act exactly as their name suggests, by “raising the alarm” of innate immunity after the exposure of airway epithelium to environmental triggers such as allergens or viruses [15].

The most common cause of asthma exacerbation is viral infections [22]. Accordingly, it is no surprise that asthma is more likely to exacerbate and is harder to control in patients with CRSwNP [23]. One of the hallmarks of nasal polyposis is epithelial barrier dysfunction since the expression of tight junction and cell adhesion proteins is significantly thwarted, compared to healthy individuals, as a direct result of IL-5 on airway epithelial cells expressing the IL-5 receptor (IL-5R) [24,25]. The weakened epithelium acts like fertile soil to facilitate the alteration of the local microbiome of the sinonasal mucosa, eventually leading to persistent inflammation [1]. The compromised immune response of patients with CRSwNP is evident since they are more easily colonized by bacteria such as *Staphylococcus Aureus* and are also more prone to viral infections, which further activate the previously mentioned adaptive and immune response systems, acting like kindling in the flame of T2 inflammation [26]. Moreover, the Th1/Th17 response, which is essential in the defense of the host to viral infections, is dysregulated in T2-high inflammatory diseases and instead, T2 biomarkers are increased after exposure to viruses such as eosinophils, IL-4, IL-5 and IL-13 in nasal fluids and bronchoalveolar lavage (BAL) [27]. This aberrant T2 response to viral infections and the presence of T2 cytokines in turn leads to a less potent production of type I interferons which normally have anti-viral effects [28]. This immune dysregulation can also explain the finding of increased viral replication in asthmatics compared to healthy subjects in an in vivo study following experimental rhinovirus infection [27].

Growing evidence is showing that CRSwNP, especially in association with severe asthma, is also linked to mucus plugs and bronchiectasis. As many as 45% of patients with bronchiectasis have been diagnosed with CRSwNP. On the other hand, bronchiectasis is more common in patients with CRSwNP compared to asthmatics [29]. This corroborates the hypothesis of the “united airways diseases”, which in the context of T2-high inflammation, highlights the role of eosinophils and IL-5 in promoting and sustaining upper and lower airways manifestation. The presence of bronchiectasis along with severe asthma has been linked with more frequent exacerbations, possibly in the context of a more prone-to-infections respiratory system due to the presence of bronchiectasis [30]. There is even the possibility that T2-high inflammation is the actual cause that leads to bronchiectasis development in the first place and the real overlap of diseases such as severe asthma, CRSwNP and bronchiectasis may be much higher than estimated. These findings suggest that in newly diagnosed patients with severe asthma, especially if it coexists with CRSwNP or needs high OCS doses to maintain control, a high-resolution computed tomography (HRCT) of the lungs is an almost mandatory step. Consequently, experts can personalize treatment based on existing comorbidities before initiating treatment with an mAb CT score and ameliorate patient outcomes. [31].

A distinct phenotype of coexisting asthma and nasal polyposis is aspirin-exacerbated respiratory disease (AERD). It is marked by CRSwNP, difficult-to-control asthma and adverse respiratory reactions to medications such as aspirin or non-steroid anti-inflammatory drugs (NSAIDs) which inhibit cyclooxygenase-1 (COX-1) [32]. Eosinophils and mast cells play a detrimental role in AERD pathophysiology by releasing inflammatory mediators, though unlike asthma where eosinophils are in the spotlight, mast cells are thought to play a more important role in the pathogenesis of CRSwNP by secreting numerous pro-inflammatory molecules. The most prominent are prostaglandin D2 (PGD_2_), prostaglandin F2alpha (PGF_2α_) and cysteinyl leukotrienes such as LTE_4_ [33,34].

PGD_2_ is an inflammatory mediator which can bind to the chemoattractant receptor-homologous molecule (CRTH2) found on the surface of T_H_2 cells, ILC2, eosinophils and basophils, activating and recruiting them to the airways. Additionally, it can bind to the DP1 receptor, also leading to chemotaxis of pro-inflammatory cells and causing nasal edema by inducing vasodilation [33]. PGF_2α_ is another eicosanoid produced by eosinophils, mast cells and possibly epithelial cells of the respiratory tract. It also acts as an agonist for the CRTH2 receptor [35]. Furthermore, it has been known for almost 50 years that inhalation of PGF_2α_ can induce bronchoconstriction even in healthy individuals, but asthmatics are far more sensitive and respond with more severe flow impairment, implying that in patients with comorbid CRSwNP and bronchial asthma the excess production of PGF_2α_ can worsen the symptoms of asthma [36]. Last but not least, LTE_4_ is another eicosanoid product which is produced mainly from eosinophils and mast cells. IL-5 plays a crucial role in regulating its production since inhibition of IL-5 signal through anti-IL-5 or anti-IL-5R biologics drastically reduces its levels. LTE_4,_ in turn, can upregulate the production of PGD_2_ and PGF_2α_, therefore inhibition of IL-5 acts beneficially for two crucial inflammatory mediators [37,38].

IL-5, as previously mentioned, is the most crucial cytokine in the pathogenesis of T2-high inflammation. Its effect is not limited to asthma since it can also induce the production of pro-inflammatory eicosanoids associated with AERD. In patients with AERD, the expression of IL-5R on the surface of eosinophils and mast cells is substantially increased, facilitating the T2 signaling process [39]. Additionally, IL-5 has recently been implicated in the process of weaking the epithelial barrier among airway epithelial cells (AEC). AEC have a fully functional IL-5R on their surface and it is plausible to discuss whether the concentration of IL-5 can affect the cell-to-cell adhesion of AEC [24]. For all these reasons, blocking IL-5 signaling is currently the most dominant weapon in combatting T2-high inflammation. An indirect marker of success for anti-IL-5 treatment is the higher expression of the CRTH2 receptor on the surface of eosinophils and basophils. Since deprivation of IL-5 reduces the levels of all three aforementioned eicosanoids, the receptor remains on the surface of the granulocytes instead of being internalized to proceed with molecular signaling [35,40,41].

IgE is the oldest target of biologic agents for asthma, with almost 20 years of experience with the anti-IgE mAb omalizumab. The role of IgE is also important in the coexistence of asthma and nasal polyps, especially in an allergic background, for instance, in patients with aeroallergen sensitization or other microbial antigens [42]. It is also considered a prominent T2 inflammation biomarker since elevated levels of IgE can increase the inflammatory response from mast cells and basophils. The ultimate goal of anti-IgE treatment is to reduce the binding of basophils with IgE and eventually replace them with new basophils, which will not be sensitized by IgE and thus not driven into an aberrant inflammatory response. The same applies for lung mast cells, which, if not sensitized by IgE, do not degranulate and do not promote allergic reactions [43]. Additionally, not only do the serum levels of IgE positively correlate with the severity of mucosal disease in patients with asthma and nasal CRSwNP, but elevated levels of IgE in sinonasal tissue is also a marker of a higher disease burden, highlighting its key role [44,45]. The role of IgE in eosinophilic centered inflammation is not yet elucidated, however, literature suggests that preventing the binding between IgE and the CD23 receptor on B-cells can halt the production of IL-5 and, therefore, subside the T2 inflammatory cascade around eosinophils [46].

It is clear that the link between these two respiratory disorders is strong, and in the era of personalized medicine, health experts should aim to optimize treatment for both asthma and CRSwNP. The shared molecular pathway of T2 inflammation is beneficial for targeted therapy with mAbs against pivotal cytokines, such as IL-4, IL-5 and IL-13 in both diseases. Initial results have shown remarkable success, and as our arsenal is growing, it is tempting to endotype asthma and CRSwNP when they coexist in order to choose the right biologic agent not for one, but for two health conditions, simultaneously. In the near future, we may even be in a position to target T2 inflammation in multiple steps with a combination of biologics, although, at the moment, no trials have been designed to test this hypothesis.

## 3. Mepolizumab

In most RCTs regarding the efficacy of mepolizumab in severe eosinophilic asthma, NP was a comorbidity and patients were evaluated as a total population for the primary outcome, that being, the rate of exacerbations in the majority of them. In a large RCT—the DREAM study—62/616 patients (10%) had comorbid NP (the percentage ranging from 7–14% in the four subgroups receiving IV mepolizumab 75 mg, 250 mg, 750 mg and placebo) [47]. In the MENSA study, 14–17% of included patients had NP [12]. A decrease in the number of clinically significant exacerbations in all mepolizumab-treated subjects compared with placebo was observed in both the DREAM and the MENSA studies, irrespective of the presence of nasal polyps at baseline. In the COLUMBA study that assessed the long-term safety and efficacy of mepolizumab, 24/347 patients (7%) presented with NP [48]. In the OSMO study which included 145 patients with uncontrolled severe eosinophilic asthma that were directly switched from omalizumab to mepolizumab and significantly improved asthma control and exacerbation rate, 20 patients (14%) had comorbid NP [49]. A post-hoc analysis of the MUSCA study and a meta-analysis of the MUSCA and MENSA studies assessed the change in HRQoL in patients with severe eosinophilic asthma, either with or without NP, treated with mepolizumab.

In the MUSCA study, 19% of patients presented with NP at baseline [50]. The mean SNOT-22 scores at baseline were 43.6 and 31.1 for patients with and without NP, respectively, indicating greater disease burden among patients with NP. Among patients with NP, mepolizumab reduced the mean SNOT-22 score by 13.7, compared to a reduction of 1.9 for placebo from baseline to week 24. This treatment difference of −11.8 exceeded the MCID, indicating a clinically meaningful improvement. In patients with severe eosinophilic asthma without NP, mepolizumab reduced (sinonasal outcome test) SNOT-22 by 4.9, demonstrating a less significant impact in the absence of NP [51].

In the meta-analysis of the MUSCA and MENSA studies, 166 out of 936 patients (18%) presented with NP at baseline. Mepolizumab reduced the annual exacerbation rate in patients with severe eosinophilic asthma versus placebo regardless of NP status, but to a greater degree in those with NP (80%) than without NP (49%). Of note, patients with NP had a higher baseline blood eosinophil count [51].

The phase 3 study, SYNAPSE, was a 52-week, randomized, double-blind, parallel group study that assessed the efficacy and safety of mepolizumab compared to placebo in patients with recurrent severe bilateral NP. Such patients were defined as those with an average nasal obstruction visual analogue scale (VAS) symptom score >5 and an endoscopic score of at least 5 out of a maximum score of 8, with a minimum score of 2 in each nasal cavity despite treatment with standard of care. Patients also must have had a history of at least one prior surgery for NP in the previous 10 years. Mepolizumab demonstrated significant improvements in the total endoscopic nasal polyp score at week 52 and in nasal obstruction VAS score during weeks 49–52, compared to placebo. Moreover, the time to the first actual nasal surgery up to week 52—a key secondary endpoint—was also statistically significantly longer with mepolizumab vs. placebo showing a reduction of 57% [52].

In a real-life study from Italy, assessing the efficacy and safety of mepolizumab treatment after 1 year, 81/138 patients (59%) had NP, thus being the most frequent comorbidity of this population [53]. In the Australian Mepolizumab Registry, 309 patients with severe eosinophilic asthma were treated with mepolizumab and 34% of them presented with NP. Super-responders, those patients at the upper 25% of ACQ-responses from baseline, or those who achieved well-controlled asthma (ACQ-5 < 1) after 6 months of treatment were more likely to have a diagnosis of NP (48% vs. 30%, *p* = 0.0350) [54]. In a real-world study from the UK including 99 patients with severe eosinophilic asthma treated with mepolizumab, NP was a comorbidity in 46/99 (46.5%) patients. NP was associated with responder and super-responder status, the former defined as >50% reduction in exacerbations or >50% reduction in OCS and the latter as exacerbation-free and off-systemic corticosteroids for at 1 year [55]. In another real-world study from Belgium, 73/116 patients treated with mepolizumab (63%) had NP [56]. In the French real-world study including 146 mepolizumab-treated patients, 38.7% of them had NP [57], while the respective percentage in a real-world study from the Netherlands was 30.8% (24/78 patients) [58].

An Italian real-word study recruited 44 severe asthmatics with eosinophilic asthma and CRSwNP who received mepolizumab and followed their course for 12 months. SNOT-22 and total endoscopic nasal polyp score (TENPS) were decreased by a statistically important margin already at 6 months and persisted until the end of the follow up period, while the ACT significantly increased over the same period. FEV_1_ was also increased at 12 months and OCS intake and blood eosinophils were drastically reduced [59]. Furthermore, in a study including 105 patients with severe recurrent NP requiring surgery, 54 patients received mepolizumab 750 mg IV every 4 weeks for a total of 6 doses and 51 patients received placebo. A history of asthma was present in 81% of the mepolizumab-treated and 75% of placebo-treated patients, and all patients with asthma had mild-to-moderate disease. The primary endpoint was a composite one based upon endoscopic nasal polyp score and nasal polyposis severity visual analog scale (VAS) score. Mepolizumab was associated with a significant reduction in the need for surgery and improved nasal polyp scores and symptoms compared to placebo [60].

Finally, the beneficial results of mepolizumab treatment have been demonstrated in a retrospective study including 32 patients, with half of them having concurrent severe asthma and bronchiectasis, which, as previously mentioned, often coexist along with CRSwNP. Treatment with mepolizumab managed to reduce the asthma exacerbation rate, the number of blood eosinophils and the OCS consumption, with better symptom control, nonetheless. The interesting point of this study is that asthma control was achieved regardless of the severity of bronchiectasis. This could possibly signal the effectiveness of mepolizumab treatment in patients with the “united airways diseases” phenotype [61].

## 4. Benralizumab

Among baseline characteristics in patients with severe eosinophilic asthma that could predict the response to treatment with benralizumab, NP was found to be one of the most important. In a study including patients from the SIROCCO and CALIMA phase III studies, it was shown that NP were associated with a greater reduction in the annual exacerbation rate and greater improvement in pre-bronchodilator FEV1 for those with >300 eosinophils/µL compared to those with severe asthma but without NP. Interestingly, NP could also predict a better response to benralizumab in reducing the exacerbation rate even in patients with <300 eosinophils/µL [62].

A substudy of the ANDHI study included 153 patients with severe asthma and NP (23.3% of the overall population) that received benralizumab (*n* = 96) or placebo (*n* = 57) for 24 weeks. Treatment with benralizumab significantly improved symptoms of NP, as assessed by SNOT-22, compared to placebo for those with a high baseline SNOT-22 score (>30). The percentage of patients with clinically meaningful improvements in SNOT-22 (at least 8.9—so called “responders”) from baseline to week 24 was greater for benralizumab compared with placebo (71.3% vs. 45.5%). This effect was even more obvious for those with a high baseline SNOT-22 score (79.7% vs. 48.8%). Moreover, a 69% reduction in the annualized exacerbation rate was observed compared to placebo as well as improvements in SGRQ, FEV1 and ACQ-6. The frequency of adverse events was similar in the two groups, and the most commonly (frequency ≥ 5% in the benralizumab group) reported, at a higher frequency in benralizumab vs. placebo, included headache, sinusitis, pyrexia and influenza [63]. The results from this substudy demonstrate the efficacy and safety of benralizumab for patients with severe asthma and NP, indicating that improvements were evident for symptoms related to NP as well as for outcomes related to asthma such as lung function and HRQoL. These improvements in asthma outcomes were better when compared to the overall ANDHI population, thus reducing the annual exacerbation rate (69% vs. 49%), SGRQ total score (−16.7 vs. −8.11), ACQ-6 (−0.88 vs.−0.46) and improving lung function (FEV1 +0.32 L vs. +0.16 L). It should also be noted that the patients included in the substudy had a greater median blood eosinophil count at baseline (510 cells/μL vs. 390 cells/μL, respectively) compared to the overall patient population [64].

In a real-life study from Italy including 59 patients with severe asthma, treatment with benralizumab for at least 6 months resulted in significant improvements in asthma-related outcomes such as ACT score, exacerbation rate, hospitalization rate and OCS use. Nasal polyposis was present in 34/59 of them, and these patients had significantly improved SNOT-22 score and reported anosmia. Compared to those without NP, a similar reduction in exacerbation rate was noted, but a greater reduction in steroid dependence and a greater improvement in lung function were demonstrated in those with severe asthma and NP [65].

In the larger real-world study from Italy, 137 patients with late-onset asthma were treated with benralizumab for 24 weeks. Among them, 79 (57.7%) presented with CRwNP. Patients with comorbid CRwNP presented with a significant improvement in SNOT-22 score, but significant improvements were also observed in ACT score, OCS dosage, FEV1 % and FEV1 (L), compared to patients without NP. Of note, the FeNO value was significantly higher in severe eosinophilic asthma with CRSwNP compared to severe eosinophilic asthma without NP (55 vs. 37.9 ppb, respectively). The study confirmed the greater benefit of benralizumab in asthma control and lung function when asthma and NP coexist. This is possibly explained by a higher degree of type 2 inflammation and eosinophilic infiltration observed in such patients [66].

A retrospective review of 23 patients with severe eosinophilic asthma and CRSwNP revealed that treatment with benralizumab for 4 months was associated with significant improvements in ACT score, FEV1 and SNOT-22 score. Despite a non-significant improvement in endoscopic polyp score, a complete disappearance of NP was observed in 36% of the patients. It was noted that the improvement in ACT scores was associated with a reduction in SNOT-22 scores and endoscopic polyp scores. Those patients who demonstrated a reduction in endoscopic polyp scores presented with higher baseline eosinophil numbers [67]. Similar improvements in SNOT-22 score and endoscopic polyp nasal score were detected in 10 patients with severe eosinophilic allergic asthma with CRSwNP after 24 weeks of treatment with benralizumab. This study showed the positive impact of benralizumab on NP in atopic patients with severe eosinophilic asthma [68].

Another real-world study from Japan including 17 patients with severe eosinophilic asthma that were treated with benralizumab for 1 year showed that the presence of CRSwNP could predict a rapid response to treatment within 4 weeks for ACQ-5, AQLQ and FEV1. On the other hand, a lack of NP (CRSsNP) was associated with a longer time to respond to benralizumab. It should be noted that eosinophils rapidly decreased to zero in both severe asthma with CRSwNP and CRSsNP groups after 4 weeks of treatment. These findings indirectly suggest that the time period to assess the response to benralizumab might be longer in patients with severe eosinophilic asthma without NP [69].

In a randomized double-blind study, 24 patients with severe NP defined by endoscopic grade >5 out of 8 and with a history of previous surgical or endoscopic polypectomy were allocated to receive benralizumab or placebo for 20 weeks. All but two of the patients had comorbid mild-to-moderate asthma. Benralizumab-treated patients showed a significant reduction in endoscopic NP score compared to baseline but a non-significant one compared to placebo. Moreover, benralizumab also improved CT score compared to baseline but not compared to placebo. All patients treated with benralizumab had an improved SNOT-22 score. Noteworthy, the ratio of blood eosinophil count to allergen skin test positivity correlated with polyp reduction, thus being a good predictor of the response to benralizumab. Accordingly, patients with a blood eosinophil count greater than 700/μL and those with negative skin prick testing all improved with benralizumab [70].

## 5. Omalizumab

Two Phase 3 trials (POLYP-1 and POLYP-2) assessed the safety and efficacy of omalizumab in CRSwNP. In these studies, 138 and 127 adults with CRSwNP that presented with an inadequate response to intranasal corticosteroids were randomized (1:1) to omalizumab or placebo and intranasal mometasone for 24 weeks. Both studies met both the coprimary end points, i.e., change from baseline to week 24 in Nasal Polyp Score and Nasal Congestion Score. Moreover, secondary endpoints—SNOT-22 score, University of Pennsylvania Smell Identification Test score, sense of smell, postnasal drip and runny nose—were also significantly improved for omalizumab versus placebo. Omalizumab was well-tolerated [71].

In a subgroup analysis of the replicate POLYP 1 and POLYP 2 trials, omalizumab efficacy was favored over placebo in patients with a blood eosinophil count >300 and ≤300 cells/μL, with or without previous sinonasal surgery, concomitant asthma and aspirin sensitivity, indicating broad efficacy regardless of underlying patient characteristics [72].

An open-label 52-week extension study evaluated the continued efficacy, safety and durability of the response to omalizumab in adults with CRSwNP who completed POLYP 1 or POLYP 2. Patients who continued omalizumab experienced further improvements, while patients who switched from placebo to omalizumab experienced favorable responses across end-points that were similar to POLYP 1 and POLYP 2. Discontinuation of omalizumab resulted in gradually worsened scores over the 24-week follow-up, but these still remained improved from pretreatment levels. No safety issues were raised [73].

A systematic review and meta-analysis of four randomized controlled trials including 303 patients aimed to assess the efficacy and safety of omalizumab for CRSwNP. Omalizumab significantly improved nasal polyps score, nasal congestion, SNOT-22 and total nasal symptom score and reduced the need for surgery in adults with moderate-to-severe CRSwNP, and it was safe and well tolerated [74].

A real-life study from Italy included 123 severe asthma patients of whom 17 (13.8%) had comorbid CRSwNP. There was no significant difference in ACQ, FEV1 and annual exacerbation rate between those with CRSwNP and those without NP. However, the proportion of patients who achieved an improvement in all three outcomes was numerically greater in the CRSwNP group (35.7% vs. 23.0%) [75]. In another real-life study with 24 patients with severe allergic asthma and CRSwNP, a 6-month treatment with omalizumab resulted in significant improvements in asthma outcomes (symptoms, rescue medication, ACT, lung function, exacerbations) and sinonasal symptoms but not on nasal polyp endoscopic score [76].

## 6. Dupilumab

LIBERTY SINUS-24 (24-week duration) and LIBERTY SINUS-52 (52-week duration) were two multinational, multicenter, randomized, double-blind, placebo-controlled, parallel-group studies assessing dupilumab added to standard of care in adults with severe CRSwNP. SINUS-24 included 276 patients, with 143 in the dupilumab group and 133 in the placebo group receiving at least one study drug dose. SINUS-52 included 448 patients, with 150 receiving at least one dose of dupilumab every 2 weeks, 145 receiving at least one dose of dupilumab every 2 weeks for 24 weeks and every 4 weeks until week 52, and 153 receiving at least one dose of placebo. In both studies dupilumab significantly improved nasal polyp score, nasal congestion or obstruction score and Lund-Mackay CT score versus placebo [77].

In a pooled analysis of the phase 3 SINUS-24 and SINUS-52 studies, 428 patients had CRSwNP and comorbid asthma. In this population, dupilumab vs. placebo improved the nasal polyp score, nasal congestion score, Lund-Mackay CT score, peak nasal inspiratory flow and SNOT-22 score. Moreover, dupilumab significantly improved FEV1 and ACQ-6 vs. placebo [78]. A further pooled analysis of the SINUS-24 and SINUS-52 studies evaluated dupilumab efficacy based upon systemic corticosteroid use and prior sinonasal surgery. It was demonstrated that dupilumab was associated with significant improvements vs. placebo in nasal polyp, nasal congestion, Lund-MacKay CT score, UPSIT and SNOT-22 scores regardless of prior sinonasal surgery or systemic corticosteroid use and also resulted in a reduced need for sinonasal surgery and steroid use and fewer steroid courses [79]. Another post-hoc analysis of the SINUS-24 and SINUS-52 studies showed that, using thresholds of clinically meaningful within-patient change from baseline on patient-reported symptoms and objective outcomes, dupilumab was associated with significantly higher responder rates compared to placebo [80].

In a large study including 724 patients with CRSwNP, the efficacy of dupilumab was evaluated according to the number of prior surgeries and the time since last surgery. Patients were randomized to placebo or dupilumab 300 mg every 2 weeks; 459/724 (63.4%) patients had ≥1 prior surgery. Compared to placebo, dupilumab significantly improved nasal polyp score, nasal congestion, Lund-Mackay CT score and SNOT-22 score in all subgroups defined by number of surgeries and by time since last surgery. However, nasal polyp score and Lund-Mackay CT score improved to a greater extent in those patients with <3 years since last surgery than patients with ≥5 years, indicating a stronger effect associated with a shorter duration from last surgery [81]. In a randomized, double-blind, placebo-controlled study, 60 adults with CRSwNP refractory to intranasal corticosteroids (including 35 with comorbid asthma) were treated with dupilumab or placebo for 16 weeks. Dupilumab vs. placebo significantly improved nasal polyp score, the Lund-Mackay CT total score, SNOT-22 and sense of smell assessed by UPSIT [82]. In a study including 35 patients with CRSwNP and comorbid asthma, dupilumab showed a significant improvement in endoscopic nasal polyp score, sense of smell, Lund-Mackay computed tomography total score and SNOT-22 score vs. placebo. Furthermore, dupilumab also produced significant improvements in asthma related outcomes, thus FEV1, and ACQ-5 vs. placebo [83].

## 7. Discussion

There is no doubt that the era of biologics has dramatically changed the management of severe asthma, mainly by reducing the rate of exacerbations and OCS use but also by improving lung function, asthma control and quality of life. CRSwNP is a common comorbidity of asthma and, quite often, of the severe asthma. Most studies on biologics have demonstrated that severe asthma with nasal polyps is an asthma phenotype that is more likely to respond to the biologic treatment. This advanced response applies to exacerbation rate reduction, OCS reduction and lung function improvement.

Patients with severe asthma and comorbid nasal polyps are eligible for biologic treatment, but recently, patients with nasal polyps and comorbid mild-to-moderate asthma also became eligible. In the case of severe asthma with comorbid nasal polyps, the precise definition of the asthmatic phenotype may guide the decision for the choice of biologic therapy. In the case of severe CRSwNP and mild-to-moderate asthma, the question is, which should be the choice of biologic therapy, or even better, which should be the first choice? Is it worth phenotyping mild-to-moderate asthma again to understand the underlying mechanism which is likely to be the same as that responsible for severe CRSwNP? Th2-predominant inflammation associated with high levels of IL-5 and eosinophils is evident in the majority of patients with severe asthma and nasal polyps but definitely not in all of them. Moreover, there are no randomized controlled studies comparing the effect of different mAbs in patients with severe asthma and comorbid CRSwNP, and accordingly, the main question remains, which mAb should be preferred in such a patient? Some systematic reviews have attempted to shed light on the answer to this question.

A systematic review and network meta-analysis regarding the efficacy of biologics in CRSwNP included nine RCTs with 1190 patients comparing three different biologics (dupilumab, omalizumab and mepolizumab) and placebo. In terms of nasal polyp score, SNOT-22, University of Pennsylvania Smell Identification Test (UPSIT) score and nasal congestion score, dupilumab had the best efficacy. Omalizumab ranked second in efficacy in terms of SNOT-22, UPSIT and nasal congestion score, while mepolizumab ranked second in efficacy in terms of nasal polyp score but had the highest risk of adverse events [84].

A Cochrane database review assessed the effects of biologics on the treatment of chronic rhinosinusitis and included 10 RCTs with 1262 patients. Dupilumab improved SNOT-22 score compared to placebo and resulted in a reduction in disease severity measured by VAS and in the number of serious adverse events. Mepolizumab may improve SNOT-22 score but it is unknown if there was a difference in disease severity or the number of serious adverse events. Furthermore, omalizumab probably improved SNOT-22 score compared to placebo but it also is uncertain if there was a difference in the number of serious adverse events, while no evidence was presented regarding the effect of omalizumab on disease severity [85].

A systematic review following the EAACI guidelines evaluated the efficacy and safety of biologicals for CRSwNP compared with standard care. Dupilumab reduced the need for surgery and OCS use (RR 0.28) and definitely improved smell (evaluated using UPSIT score) and quality of life (evaluated using SNOT-22 score) with fewer treatment-related adverse events. Omalizumab also reduced the need for surgery (RR 0.85), decreased OCS use (RR 0.38) and improved smell and SNOT-22, but increased treatment-related adverse events. It is unclear if mepolizumab actually reduced the need for surgery or improved QoL while it also demonstrated an increase in treatment-related adverse events. The evidence for reslizumab is also very uncertain [86]. An indirect comparison of the efficacy of dupilumab and omalizumab, including two studies of dupilumab (SINUS-24 and SINUS-52) and two studies of omaluzumab (POLYP-1 and POLYP-2), found that dupilumab was associated with greater improvements from baseline to week 24 vs. omalizumab with regards to nasal polyp score, nasal congestion, loss of smell and total symptom score, whereas SNOT-22 improvement was greater for dupilumab, although not significantly, compared to omalizumab [87].

Despite the appreciation that indirect comparisons are not safe to exclude definite conclusions it seems that changes in nasal polyp score are greater in absolute or percentage terms for dupilumab than for omalizumab and mepolizumab. This is possibly attributed to the mechanism of action. Dupilumab acts at a higher level of the type 2 inflammatory cascade by blocking signaling of IL-4 and IL-13. Real-life studies have shown similar trends. In a real-life setting, 28 patients with CRSwNP were treated with mAbs (mepolizumab, omalizumab and benralizumab), and all but one had concomitant asthma (but not necessarily severe). Treatment with mepolizumab was associated with a better success rate (78.9%) followed by omalizumab (50%) and benralizumab (50%) but the latter was with few cases [88]. However, no significant change in polyp score was found in the mepolizumab-treated group, which is in contrast to studies by Bachert et al. [60] and Gevaert et al. [89]. This might be attributed to the intravenous administration of a higher mepolizumab dose of 750 mg in the latter two studies. The same applied to omalizumab, with an increase in the overall average polyp score. Moreover, no predictive biomarkers for a response to treatment could be identified. Furthermore, improvements in nasal symptoms as assessed by SNOT-22 but no improvement in endoscopic findings as evaluated by the nasal polyp score were demonstrated in another observational study from Italy which included 33 severe asthmatics with comorbid NP who were treated with omalizumab, mepolizumab and benralizumab for 52 weeks [90].

For severe asthma, the choice among mAbs is guided by the asthma phenotype, and the option of switching from one to another exists in the case of suboptimal a response. Comorbid CRSwNP represents a unique phenotype that enhances biological effectiveness.

In the case of CRSwNP with or without asthma (not necessarily severe), there is neither clear evidence nor biomarkers that could guide the initial biologic choice.Additionally, there are no switching studies from one biologic to another and, as previously mentioned, only indirect comparisons exist among them, making the clinician‘s decision more difficult. A recent study evaluated the efficacy of benralizumab in patients with CRSwNP, where it managed to improve nasal polyp score and nasal blockage score, but this improvement was not demonstrated in patients with BMI >30, patients with fewer than 2 nasal polyp surgeries, patients without comorbid asthma and those with fewer than 560/μL blood eosinophils [91]. This study indicated that a combination of phenotypic characteristics and biomarkers may aid the choice of biologic. As more real-life studies accumulate, we will gather more data regarding the differences in effect on the various outcomes in the population of patients with severe CRSwNP. Until then, and while we try to get more information on the underlying pathophysiology of CRSwNP, an in-depth phenotyping of each patient’s comorbid asthma along with the search for other type 2 comorbidities, such as atopic dermatitis, allergic rhinitis and urticaria, might help us to choose the proper biologic. Meanwhile, other factors such as frequency and/or route of administration and patients’ input should also be taken in consideration.

## 8. Conclusions

CRSwNP is a chronic inflammatory disease of the nasal passage linings or sinuses often combined with severe asthma. This combination of severe asthma with CRSwNP presents a unique phenotype with a better response to biologic treatment which is evident in the reduction of asthma exacerbations, use of maintenance steroids and improvement in asthma control, lung function and asthma-related quality of life. CRSwNP without asthma or with mild-to-moderate asthma may be treated effectively with biologics, but the choice of which one is a matter of investigation. The phenotype of mild-to-moderate asthma may aid such decisions. Further studies involving nasal biopsies and a multidisciplinary approach, including the asthma and ENT specialist, may assist in defining clinical characteristics and/or biomarkers for the best choice of biologic treatment in CRSwNP.

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
