# Peer review of "Biological Therapy of Severe Asthma and Nasal Polyps"

_jpm, 2022, doi:10.3390/jpm12060976_

Round 1

Reviewer 1 Report

I’ve read carefully this narrative review presented by Bakakos and colleagues about asthma and nasal polyposis. This topic is of particular interest among respiratory medicine physicians, ENT specialists and allergologists, especially with the availability of biological therapy which can be used for the treatment of T2-high manifestations of both upper and lower airways.

The review is overall well written. Notwithstanding, some issues need to be addressed:

-Title: The current title is somewhat too generic. Since this review focuses on the biological therapy of asthma and nasal polyps I suggest changing the title as: “Biological Therapy of Severe Asthma and Nasal Polyps”.

-Abbreviations: CRSwNP (Chronic Rhinosinusitis with Nasal Polyps) and CRSsNP (Chronic Rhinosinusitis without Nasal Polyps) are the most common in literature and should be used throughout the text.

-Introduction: Current treatment for NP includes biological therapies in eligible patients. Hence, these should be added in this paragraph. Furthermore, a bit of information about severe asthma prevalence, its socioeconomic burden and the importance of the association between severe asthma and CRSwNP should be introduced, focusing on the synergistic effects of both diseases on symptoms, exacerbations and OCS consumption and on the possibility to treat both with monoclonal antibodies.

-The link between asthma and nasal polyps:

1- Please consider to change the title of this paragraph in “The link between severe asthma and nasal polyps”

2- Growing evidence is showing that CRSwNP, especially in association with severe asthma, is also linked to mucus plugs and bronchiectasis. This corroborates the hypothesis of the“united airways diseases”, which in the context of a type 2-high inflammation, highlight the eosinophils role in promoting and sustaining upper and lower airways manifestation. Thus, severe asthma biological therapy could limit airway damage phenomena via blocking immune cells and cytokines associated with type 2-high inflammation. In order to better address this aspect, I suggest to add the following references:

  • doi:10.1097/ACI.0000000000000492
  • doi:10.2147/JAA.S332245
  • doi:10.1016/j.jaip.2021.04.054
  • doi:10.1016/j.rmed.2021.106491

-Mepolizumab: I suggest to expand the references concerning this section adding the following article: doi:10.1177/17534666211009398

-Discussion: This section needs to be improved. The discussion is mainly developed by presenting studies that indirectly compare biological therapy effectiveness in CRSwNP and severe asthma outcomes. I suggest rewriting the discussion, focusing only on the most relevant indirect network meta-analysis comparisons and articulating a more critical discussion on the overall use of monoclonal antibodies for CRSwNP and severe asthma alone or when the two diseases are both co-present.

Author Response

I’ve read carefully this narrative review presented by Bakakos and colleagues about asthma and nasal polyposis. This topic is of particular interest among respiratory medicine physicians, ENT specialists and allergologists, especially with the availability of biological therapy which can be used for the treatment of T2-high manifestations of both upper and lower airways.

The review is overall well written. Notwithstanding, some issues need to be addressed:

-Title: The current title is somewhat too generic. Since this review focuses on the biological therapy of asthma and nasal polyps I suggest changing the title as: “Biological Therapy of Severe Asthma and Nasal Polyps”.

Response: The authors agree with the comment and we have proceeded in changing the title according to the reviewer’s recommendation.

-Abbreviations: CRSwNP (Chronic Rhinosinusitis with Nasal Polyps) and CRSsNP (Chronic Rhinosinusitis without Nasal Polyps) are the most common in literature and should be used throughout the text.

Response: The reviewer is right; all abbreviations have been meticulously checked, formatted according to the suggestions and are highlighted in red.

-Introduction: Current treatment for NP includes biological therapies in eligible patients. Hence, these should be added in this paragraph. Furthermore, a bit of information about severe asthma prevalence, its socioeconomic burden and the importance of the association between severe asthma and CRSwNP should be introduced, focusing on the synergistic effects of both diseases on symptoms, exacerbations and OCS consumption and on the possibility to treat both with monoclonal antibodies.

Response: The reviewer is right; we have improved the introduction section and inserted the information requested. Our changes are highlighted in red. We thank the reviewer for his suggestions.

-The link between asthma and nasal polyps:

1- Please consider to change the title of this paragraph in “The link between severe asthma and nasal polyps”

Response: The reviewer is right; we have changed the title according to the suggestion.

2- Growing evidence is showing that CRSwNP, especially in association with severe asthma, is also linked to mucus plugs and bronchiectasis. This corroborates the hypothesis of the “united airways diseases”, which in the context of a type 2-high inflammation, highlight the eosinophils role in promoting and sustaining upper and lower airways manifestation. Thus, severe asthma biological therapy could limit airway damage phenomena via blocking immune cells and cytokines associated with type 2-high inflammation. In order to better address this aspect, I suggest to add the following references:

  • doi:10.1097/ACI.0000000000000492
  • doi:10.2147/JAA.S332245
  • doi:10.1016/j.jaip.2021.04.054
  • doi:10.1016/j.rmed.2021.106491

Response: The reviewer is right; a paragraph in red color has been added in the chapter including the information from the manuscripts suggested. The last suggested manuscript “doi:10.1016/j.rmed.2021.106491” has been added to the Mepolizumab section since it refers to the effectiveness of the mAb. We hope the reviewer agrees with our choice.

-Mepolizumab: I suggest to expand the references concerning this section adding the following article: doi:10.1177/17534666211009398

Response: The reviewer is right; the suggested manuscript’s data has been implemented in the “Mepolizumab” chapter with red color.

-Discussion: This section needs to be improved. The discussion is mainly developed by presenting studies that indirectly compare biological therapy effectiveness in CRSwNP and severe asthma outcomes. I suggest rewriting the discussion, focusing only on the most relevant indirect network meta-analysis comparisons and articulating a more critical discussion on the overall use of monoclonal antibodies for CRSwNP and severe asthma alone or when the two diseases are both co-present.

Response: The reviewer is right, the discussion section has been rewritten and reorganized taking into consideration the reviewer’s suggestions.

Reviewer 2 Report

It is an excellent exhaustive review on the pathogenesis of chronic rhinositusitis and its potential treatment with different monoclonal antibodies. Authors addressed the main issues raised, and the text is clear. It is a very interesting and well written manuscript.

Author Response

We would like to thank the reviewer for his comments and his appreciation for our work. 

Round 2

Reviewer 1 Report

The manuscript has significantly improved, it is informative and well organized. I would suggest to add "narrative review" in the title.